# Psychosocial Health of K-12 Students Engaged in Emergency Remote Education and In-Person Schooling: A Cross-Sectional Study

**DOI:** 10.3390/ijerph18168564

**Published:** 2021-08-13

**Authors:** Daniel Acosta, Yui Fujii, Diana Joyce-Beaulieu, K. D. Jacobs, Anthony T. Maurelli, Eric J. Nelson, Sarah L. McKune

**Affiliations:** 1Department of Environmental and Global Health, College of Public Health and Health Professions, University of Florida, 1225 Center Dr, Gainesville, FL 32611, USA; daniel.acosta@ufl.edu (D.A.); amaurelli@phhp.ufl.edu (A.T.M.); eric.nelson@ufl.edu (E.J.N.); 2College of Public Health and Health Professions, University of Florida, 1225 Center Dr, Gainesville, FL 32611, USA; yfujii@ufl.edu (Y.F.); kdotddot@ufl.edu (K.D.J.); 3Department of Special Education, School Psychology & Early Childhood Studies, College of Education, University of Florida, 618 S.W. 12th Street, P.O. Box 117050/1801, Norman Hall, Gainesville, FL 32611-7050, USA; djoyce@coe.ufl.edu; 4Emerging Pathogens Institute, University of Florida, 2055 Mowry Road, Gainesville, FL 32610, USA; 5Department of Pediatrics, College of Medicine, University of Florida, 1600 SW Archer Rd, Gainesville, FL 32610, USA

**Keywords:** COVID-19, students, psychosocial health, virtual learning, depression, anxiety, OCD, environment

## Abstract

As online classes became the norm in many countries as a response to the COVID-19 pandemic, the concern for child and adolescent mental health became an issue of concern. This study evaluates the differences in the psychosocial status of school children based on engagement in in-person or Emergency Remote Education (ERE) and assessed the prevalence and predictors of symptom-derived risk levels for anxiety, depression, and obsessive-compulsive disorders (OCD). Cross-sectional data were collected from students at a Florida K-12 school and their household members through an online survey conducted in October 2020 (*n* = 145). No significant difference was found between ERE and in-person learning for risk of anxiety, depression, or OCD. Prevalence of students presenting as at risk for anxiety, depression, and OCD was 42.1%, 44.8%, and 41.4%. Several student factors (e.g., child sex, school level) and parental factors (e.g., parental COVID-19 attitudes) were associated with students presenting as at risk for anxiety, depression, or OCD; child’s participation in sports was protective against all three outcomes. Participation in sports was found to be protective against risk of anxiety (aOR = 0.36, CI = 0.14–0.93), depression (aOR = 0.38, CI = 0.15–0.93), and OCD (aOR = 0.31, CI = 0.11–0.85).

## 1. Introduction

In an effort to adhere to public health guidelines during the 2020 COVID-19 pandemic, schools around the world have taken varying approaches to continue student education using virtual platforms [1]. By April 2020, 1.58 billion students from 200 countries were affected by school closures, which aimed to reduce transmission of COVID-19 [2]. These decisions triggered action among educational institutions globally-EdTech companies mobilized to partner with schools in Switzerland, and Indian universities pushed to transition fully online [3,4].

As a result of the global pandemic of COVID-19, schools around the world took various approaches to learning in the Fall of 2020, including full closures, in-person, fully virtual, and hybrid options. Among those that remained open, the various learning design models reflect different ways that education, epidemiology, and equity issues were prioritized by governing bodies. Municipalities worked on mitigating risk of disease transmission, as the psychosocial impact of school-age children being physically out of school and learning virtually was, at the time, largely unknown. While most of the various forms of virtual learning implemented as a response to COVID-19 meet the UNESCO definition of “distance education” [5], categorizing these efforts as distance education is problematic and misleading. Distance education is usually implemented through willingly planned and evidence-based pedagogy, as measured against student learning [6]. On the other hand, distance learning in the context of the COVID-19 pandemic is a coping strategy—an emergency temporary solution for unplanned disruptions of prior social order [7]. These approaches to learning are not necessarily agreed upon strategies and, in some cases, can reflect the only viable option of a narrow range of choices implemented to preserve some semblance of stability and education for students [8]. Schools in the US were grossly unprepared for mandatory transitions to online learning. Only 19% of all primary and secondary schools in the US had any experience offering classes entirely online as of the 2017–2018 school year [9]. In addition to school unpreparedness, families were not equitably positioned for a transition to online learning. A study conducted in 2018 showed that among low-income families, one in four teens lacked computers at home [10]. Consequently, virtual learning threatened to exacerbate the existing academic gaps between low and high socioeconomic groups in the US [11]. It is important to recognize that the objective of distance learning during COVID-19 was not to reconstruct an environment that provided a stable education, but an environment that provided immediate virtual access to most students during the crisis [12].

Educational experts acknowledging these issues describe the forced exodus of students from the classroom towards distance education as Emergency Remote Education (ERE) [13]. ERE differs from distance education because it highlights the contextual nature of learning remotely-it is both a response to a crisis and temporary, thus not planned or designed to be online [13]. In this study, we use data from a K-12 developmental research school, where student families were given the option of attending school 5 days per week or engaging in ERE in the Fall of 2020, to study differences in psychosocial health of children and adolescents. While there are several emerging studies regarding the negative psychosocial outcomes the pandemic has had in children and adolescents [14,15,16,17,18,19,20,21,22,23,24,25], no literature was found that examined or compared any indicator of child mental health outcomes across different ERE strategies, particularly among K-12 students. While the concept of change and uncertainty and their association with stress have been thoroughly studied within psychology [26,27,28,29], given the unprecedented nature of the pandemic and associated school closures, more research is needed to understand the psychosocial impact of ERE on children.

Our objective was to test for differences in risk of psychosocial outcomes in two distinct environments, between students attending school in-person and those in engaged in ERE. We also describe the prevalence of students presenting as at risk for anxiety, depression, and obsessive-compulsive disorder (OCD), and identify other socio-demographic and behavior-based factors associated with those outcomes.

## 2. Materials and Methods

### 2.1. Study Design

Data presented here were collected as part of a larger multi-disciplinary research project that began in March 2020 and aimed to identify the determinants of COVID-19 infection and transmission in Florida [19]. As part of those efforts, a prospective cohort study was launched in April 2020 to increase understanding of the role that children played in the spread of COVID-19 infection and the impacts of the COVID-19 pandemic on school-aged children and their families. The cohort study was designed to collect data on children and their families over 12 months, including tests for active COVID-19 infection, serologic tests for past exposure to the virus, and an online household survey, to explore the situation of children and their families, first during the initial lockdown (April 2020) and then during six- and 12-month follow-up. Laboratory data reflecting infection and antibody status of children and their families will be published at completion of the cohort study. Data presented here come from survey data, collected in October 2020, when a significant number of children were engaged in ERE.

### 2.2. Recruitment and Participation

Parents or guardians of existing participants (from April 2020) and all other currently enrolled students were invited by email to have their child participate in the October 2020, six-month follow-up phase of the study. No compensation was offered to participants. Recruitment materials included emails and a website with additional information about the study. New participants were able to consent/assent to participation online via a HIPAA compliant interface connected to REDCap (Vanderbilt University), while existing participants, previously consented, were sent a unique REDCap link that directed them to the online survey and testing appointment sign-up. Parental consent was required for all children under the age of 18, and assent was required for all children over the age of eight. Data were deidentified prior to analysis and are stored on secured servers to ensure the protection of participants (REDCap). The study was approved by the (University of Florida) Institutional Review Board, protocol [IRB202001345].

### 2.3. Instrument and Variables

The questionnaire included parental or self-report of symptoms in children associated with anxiety, depression, and OCD, as well as knowledge, attitudes, and practices (KAP) of parents related to COVID-19, socio-demographic characteristics, the current learning environment of the student (in-person or ERE), household income, parental COVID-19 vaccine attitudes, parental health related quality of life (HRQoL), parental optimism, parental risk perception, parental resilience, household food security, parental unwanted behaviors, and parental trust in institutions. These variables are described in more detail below. The questionnaire used can be found in the Appendix A.

The outcomes evaluated in this study are symptom-derived categorizations of students as no risk or at risk for anxiety, depression, and OCD. In the bi-variate analysis, we also categorize a third group, high risk, from within the population at risk. These indicators are generated from participant responses to questions about symptoms associated with each outcome, using a 5-point Likert scale. The questions used age-appropriate language, and the response options (and associated scores) were never (1), a little (2), sometimes (3), a lot (4), or always/constantly (5). When asked about a list of symptoms associated with each outcome, participants whose answer responses were all below 3 were categorized as no risk; participants who had 1 or more answer equal to 3 or above were categorized as at risk. Within those at risk, those who reported 2 or more symptoms with scores of 4 (reflecting having that symptom “a lot”) or one or more symptom score of 5 (reflecting having that symptom “always” or “constantly”), were categorized as high risk for that outcome. A team of psychologists from the university affiliated with the participating developmental research school designed the questions to gain insight into psychosocial outcomes of interest to the school.

The primary independent variable examined in this study is learning environment, a dichotomous nominal variable that categorizes student learning environment as either ERE or attending school in-person, as defined above. A few distinctions about this context are important: first, all students were issued laptops during the initial lockdown in March 2020. Thus, all students had tried ERE in the spring of 2020, when no other option was available. All children also had access to a school-issued laptop when families were choosing their learning environment. In addition, there were differences by school level in the delivery of classes among those attending school in-person: primary school students received traditional face-to-face instruction, and middle school and high school students received instruction and engaged in class work virtually through use of laptops from within their classrooms. Middle and high school students reported to the same classroom each day along with members of a “pandemic pod”. From there, they joined their individual classes—different from others in their pod—by laptop. The analysis treats those attending school in-person as a homogeneous group and does not differentiate between those receiving traditional face-to-face classes (K-5 grade) and those on campus using a laptop to connect to class (6–12 grade), as they are considered to be in the same environment (physically at the school); however, school level is included in all regression models as a covariate.

Other covariates evaluated and included in our analyses are socio-demographic characteristics, school level, parental KAP, parental unwanted behaviors, parental vaccine attitudes, parental optimism score, resilience, healthy days, student participation in sports, and flu vaccination status. Socio-demographic variables include child race, child sex, household income, parental COVID-19 related loss of income, and parental occupation (binary variable, assessing whether or not a parent worked in a medical setting). School level is described as primary (K-5), middle (6–8), and high school (9–12). For parental KAP, continuous variables were generated for each parental knowledge, parental attitudes, and parental practices. Each correct answer on the knowledge section counted as 1 point (out of 16). Attitudes and Practices where categorized either as protective or unprotective against COVID-19, with each protective attitude counted as one point towards the attitude score (out of 10), and each protective practice counted as one point towards the practice score (out of 7).

Variables assessing unwanted behaviors and trust in institutions were adapted from the World Health Organizations (WHO) survey tool and guidance to monitor knowledge, risk perceptions, preventive behaviors, and trust [30]. Unwanted behaviors reflect a set of attitudes and actions deemed counterproductive or unhealthy during a pandemic, such as consuming more alcohol, purchasing drugs that are not used as an effective treatment, or postponing vaccinations. The resilience scale, healthy days, and optimism scores used validated instruments and analysis guidelines [31,32,33]. The variable referred to as “healthy days” is derived from the “Healthy Days Core Module” of the CDC’s HRQoL [34]. Questions about parental COVID-19 vaccine attitudes were designed by the study team to understand parental perceptions about vaccinating children prior to vaccine availability for children.

Because we are interested in the learning environment and its impact on psychosocial outcomes, we present descriptive statistics for the study population by learning environment and include indication of statistical differences between these populations where they occur. Bivariate analysis was conducted between all covariates and the outcome measures (symptom derived indicators of anxiety, depression, and OCD). Those with *p*-values of less than 0.20 were carried over to regression models. For each outcome, nested regression models are estimated to investigate the relationship between learning environment and each psychosocial outcome, taking into consideration other potential confounders. In each nested regression model, the first model includes socio-demographic confounders (sex, race, school level, and income), independent of bivariate analysis; and the second model includes learning environment (ERE or in-person). In the third, fourth, and fifth models of each nested regression, we explore the role of other covariates on the specific outcomes, including those covariates that were associated with the outcome in bivariate analysis at the *p* < 0.20 level. These are grouped and include parental health attitudes and behaviors (third model); parental occupation, self-reported health, or resilience (fourth model); and child participation in sports (fifth model). We present adjusted odds ratios for potential risk factors associated with each of three outcomes: children presenting with no risk or at risk for anxiety, no risk or at risk for depression, and no risk or at risk for OCD.

A total of 181 students (15.3% of the total school population) participated in the study. A subset of 36 high school students was removed from this analysis after it was determined that they inadvertently received questionnaire language that targeted children of a lower age. Thus, the total sample evaluated here is 145 students. The study population (*n* = 145) was 49.7% White (non-Hispanic), a 24.8% Hispanic (regardless of race), 13.1% Multiracial/Other, and 12.4% Black (non-Hispanic). Female students made up 50.3% of the sample, and the distribution by school level was 22.8% high school, 31.7% middle school, and 45.5% primary school. Median income was $100,000 annually. Table 1 presents these demographic characteristic and other study variables of interest for both the full sample population as well as by learning environment (those in school in-person and ERE). Variables of interest include parental COVID-19 vaccine perceptions, children’s seasonal flu vaccination status, indicators of psychosocial distress, parental healthy days, KAP scores, unwanted behavior scores, resilience scores, and optimism scores.

## 3. Results

Most parents (71%) reported having some degree of concern about the safety of COVID-19 vaccines for children, and only 45.5% of parents indicated that they planned to have their children vaccinated. Among students’ households, 14.5% indicated a loss of income due to COVID-19, and 33.1% indicated having an adult who worked on the front line, such as doctors or nurses. 73.8% of students participating received a flu vaccine during the 2019–2020 school year. COVID-19 related parental KAP scores were high in this population, with average scores of 14.6 out of 16, 7.7 out of 10, and 6.4 out of 8, respectively. Parents reported 25 healthy days out of the past 30. Unwanted behavior, optimism, and resilience scores are all presented in Table 1.

In testing the central thesis of this study, we found no significant difference in students presenting as at risk for anxiety (*p* = 0.452), depression (*p* = 0.167), or OCD (*p* = 0.890) when those engaged in ERE (*n* = 100) were compared to those attending school in-person (*n* = 45) students. Among the total student population, 42.1% reported symptoms consistent with being at risk for anxiety, 44.8% reported symptoms consistent with being at risk for depression, and 41.4% of students reported symptoms consistent with being at risk for OCD. Overall prevalence rates of students reporting symptoms consistent with being at high risk for anxiety, depression, and OCD were 9.7%, 5.5%, and 11.0%, respectively. A full 57.2% of students were categorized as at risk for at least one of the three psychosocial outcomes, and 13.1% were categorized as at high risk for at least one of the three psychosocial outcomes.

Descriptive statistics in Table 1 are presented for the total study population as well as by learning environment, in an explicit effort to document characteristics of those students attending school in-person and those engaged in ERE. Significant differences were found by school-level (younger students were more likely to be learning at school, *p* = 0.003), parental concern about vaccine safety in children (those with parents not reporting concern were more likely to be learning at school, *p* = 0.039), and by student receipt of a 2019–2020 flu vaccine (those who did not receive a flu vaccine were more likely to be learning at home, *p* = 0.007). In addition, significant differences were found in parental COVID-19 related attitudes and practices, with parents of those learning at home reporting higher scores, indicative of more protective COVID-19 attitudes and practices (*p* = 0.000 and *p* = 0.001).

In Table 2, we present nested logistic regression models with adjusted odds ratios for factors evaluated in models estimated to understand symptom-derived risk indicators for anxiety. Being a primary or middle school student was associated with presenting as at risk for anxiety in all of the five regression models, (aOR = 6.73, 95% CI = (1.92–23.51) and aOR = 6.17, 95% CI = (1.74–21.9), respectively), when compared to high school students. A higher parental COVID-19 knowledge score was also positively associated with a child presenting as at risk for anxiety (aOR = 1.71, 95% CI = (1.17–2.48)). Children’s participation in sports was found to be protective against a child presenting as at risk for anxiety (aOR = 0.36, 95% CI = (0.14–0.93)). There was no association at the significance level of *p* = 0.05 between anxiety and race, learning environment, income, parental self-reported healthy days, parental unwanted behaviors, parental attitudes towards COVID-19, and parental attitudes towards COVID-19 vaccines.

In Table 3, we present nested regression models that estimate the effect of learning environment on students presenting as at risk for depression. In the models, we include the adjusted odds ratios for each factor evaluated. Similar to anxiety, primary and middle school students had significantly higher odds of presenting as at risk for depression in all of the five regression models, (aOR = 5.40, 95% CI = (1.56–18.66) and aOR = 6.55, 95% CI = (1.79–23.9), respectively), when compared to high school students. Being female was also positively associated with a child presenting as at risk for depression (aOR = 2.99, 95% CI = (1.27–7.03)). Race was significantly associated with being at risk for depression in the first two models, where Black children were less likely to present as at risk for depression compared to Non-Hispanic White children. However, as parental health attitudes (model 3), parental occupation (model 4), and child participation in sports (model 5) were introduced, no significant association was found. Similar to anxiety, children’s participation in sports was found to be protective against a child presenting as at risk for depression (aOR = 0.38, 95% CI = (0.15–0.93)). In the final model, there was no association at the significance level of *p* = 0.05 between depressive symptoms and race, learning environment, income, parental occupation, parental unwanted behaviors, and parental attitudes towards COVID-19.

Table 4 provides the nested logistic regression models estimating the effects of distance learning on symptom-derived risk indicators for OCD. Adjusted odds ratios are included for all factors evaluated in models. A higher COVID-19 preventive attitude score was associated with a child presenting as at risk for OCD (aOR = 1.60, 95% CI = (1.10–2.34)). Similar to anxiety and depression, children’s participation in sports was found to be protective against a child presenting as at risk for OCD (aOR = 0.31, 95% CI = (0.11–0.85)). In the final model, there was no association at the significance level of *p* = 0.05 between OCD-related symptoms and race, sex, school level, learning environment, income, parental occupation, parental unwanted behaviors, parental attitudes towards COVID-19 vaccines, parental healthy days, and resilience score.

## 4. Discussion

Much attention has been given to the range of approaches employed to continue student education through virtual platforms during the COVID-19 pandemic, as schools sought to balance the educational and psychosocial needs of children against the potential risk of infection that in-person learning might introduce. In this study, 145 children and their families were administered questionnaires, and their responses yielded important insights into the situation of K-12 students and families roughly six months after COVID-19 restrictions began in Florida.

In our sample, 68.9% of students were connecting virtually via school-issued laptops from home, while the remaining 31% were attending school in-person. Though our core research question aimed to understand the differences in symptom-derived risk levels for anxiety, depression, and OCD between these groups, our analysis found no significant difference in these psychosocial outcomes between the two groups of learners. We found no literature that compares the psychosocial outcomes of K-12 students attending in-person classes with those engaged in ERE of any type in any context similar to the global COVID-19 pandemic, making this study the first of its kind. Given the scale of effort expended by school, public health, and other decision-making officials to weigh the COVID-19 risks and benefits against the psychological risks and benefits of virtual versus in-person learning, these data were desperately needed in July and August of 2020 and should prove useful for policy makers moving forward.

Unlike learning environment, being a primary or middle school student did increase the likelihood that a student presented as at risk for anxiety and depression. Younger students might not adapt as easily to ERE as those in high school as they might be less familiarized with the technology needed for online classes. Qualitative research conducted with primary school teachers and parents in Indonesia found that limited socializing and limited communication were challenges for young students, and that longer screen time was a challenge for young students. Parents also reported a lack of discipline around learning at home and poor technology skills in their children as challenges [35]. These findings may explain some of the challenges faced by primary students in the US too. However, findings that younger children are at higher risk for poor psychosocial or mental health outcomes during the pandemic is counter to research findings conducted elsewhere, such as a study from China that found older children at higher risk for depression and anxiety [36]. Our findings also associate being female with higher risk of depression, which is consistent with other COVID-19 studies where being female is a predictor of depression, anxiety, and stress [36,37,38,39]. These findings reflect similar findings from the beginning of the pandemic, where being younger and being female were associated with increased risk of symptoms of psychosocial distress [19].

This study also found playing a sport to be protective against a student presenting as at risk for anxiety, depression, and OCD; students participating in sports were 64%, 62%, and 69% less likely than those not playing sports to present as at risk for anxiety, depression, and OCD, respectively. This is consistent with a robust literature on the benefits of behavioral activation, such as exercise, sports, or re-engagement in social activities, as treatment for anxiety and depression [40,41,42,43,44]. These results are also consistent with emerging literature on the role of physical activity in mental health in the context of the COVID-19 pandemic. A study found that implementing fitness programs among adolescents during COVID-19 lockdowns in Italy increased Regulatory Emotional Self-Efficacy scores after 8 weeks [45], and a qualitative study highlighted the role of physical activities and social connections in student-athletes’ mental health [46]. However, schools around the world continue to face many challenges to promote physical activities and physical education in the context of COVID-19 [47,48]. Furthermore, inequalities in resources and built environment can act as barriers to students from lower socio-economic levels, which is something that has to be taken into consideration by educators while planning interventions aiming to increase physical activities [46].

Participation in sports also addressed concerns of limited socializing and communication presented by parents and teachers in the qualitative study in Indonesia [35]. This finding is particularly important as studies have found that children have reduced levels of physical activity and increased sedentary behaviors since the start of the COVID-19 pandemic [49,50,51,52]. Some guidelines for reopening schools advised against the reactivation of sports, especially indoors, as likely to increase risk of transmission [53,54]. This poses a paradox for decision makers, as sports and physical activity are protective against negative psychosocial outcomes in children, but they may increase the risk of COVID-19 infection among school-aged children.

Additionally, another critical finding showed that the higher the parental knowledge scores, the more likely were children to present as at risk for anxiety. Questions to assess anxiety levels in children were created to be age-appropriate for children to self-report, and parental knowledge was assessed with each question representing a score of one and a total possible score of 16. While the finding may be counterintuitive, this relationship may be explained by recent findings. A study on public perception of COVID-19 identified that higher knowledge levels were attributed to greater uncertainty and an increased compliance with health measures [55]. Moreover, a study from Saudi Arabia found that those who consistently practiced health safety measures such as wearing masks and using hand sanitizer presented significantly higher levels of anxiety compared to those who did not [56]. While a hyper-vigilant focus on the pandemic and its risk may have increased anxiety levels in children, other explanations, such as parental education level, could also explain this association, thus further investigation is needed to accurately identify the source. A similar pattern presents for COVID-19 related attitudes. Among parents who report the most protective or conservative attitudes toward COVID prevention, children are more likely to present as at risk for OCD. These highly protective attitudes around COVID-19 may be comparable to over-involved and/or protective parenting, which has been associated with early childhood internalizing difficulties [57,58,59,60].

Finally, although parental attitudes about child COVID-19 vaccination were not associated with the psychosocial outcomes of interest, the data collected in October of 2020 regarding vaccine attitudes are important. Findings showed that 45.5% of parents planned to vaccinate their children against COVID-19, while 46.2% remained uncertain, and 8.3% planned to refuse vaccination. Parents’ major concern was vaccine safety, with 71% of parents expressing some concern or that they were very concerned about safety. When data were collected, no COVID-19 vaccines were yet approved for use in either adults or children, thus these questions were hypothetical, and parents had very little data. During the final round of data collection in April 2021, with vaccines widely available in the study area for those 16 and over, we collected vaccine attitude data again. An analysis of changes in parental attitudes about child COVID-19 vaccination in this population is forthcoming.

### Limitations

The sample for this study (145 students) is small and represents a decrease from baseline data collection where 280 students participated. Additionally, despite socio-demographic characteristics of the school reflecting those of the state, over 50% of households reported an annual income of over $100,000, which is higher than the reported median for the state in 2019 of $56,000 (2019 inflation adjusted dollars, American Community Survey 2019 5-year estimates [61]). Another limitation is that previous participants (those who participated in baseline) who were over 13 years of age inadvertently received the psychosocial questionnaire designed to engage students from 8–13 years of age, thus the questions they received used language developed for younger children. Consequently, these 36 students were removed from the study, limiting the enrollment of older children and ruling out any older child who had participated since baseline. Another limitation is that the study did not control for the variation in the educator’s proficiency in online schooling, Moreover, psychosocial distress in households might have been heightened during the period of data collection (October 2020), as it was a month marked with social and political turmoil in the United States leading up to the 2020 presidential election.

## 5. Conclusions

There was no association found between students’ learning environment and their presenting as at risk for anxiety, depression, or OCD. Instead, drivers of risk included: biological sex (being female increasing likelihood of presenting as at risk for depression), and school level (being in primary school and middle school increasing the likelihood of presenting as at risk for anxiety and depression). Higher levels of parental knowledge were associated with presenting as at risk of anxiety, and higher attitude scores were associated with presenting as at risk of OCD. Importantly, participation in sports was protective against students presenting as at risk of anxiety, depression, and OCD. These data both clarify and complicate the role that sports play in the health of children during a pandemic. More efforts are needed to fully understand how to protect and optimize the educational, mental, and physical health of children during a pandemic. Additional research is needed to understand if these patterns replicate across various populations in the US and globally. Such data would foster the allocation and prioritization of resources for interventions to support the children most vulnerable to the negative impact of the pandemic.

## Figures and Tables

**Table 1 ijerph-18-08564-t001:** Demographic Characteristics of the Study and Variables of Interest by Learning Environment and for Total Study Population.

	Is Child Attending in Person School?		All Participants *n* = 145 (Column %)
	Yes *n* = 45 (Row %)	No *n* = 100 (Row %)	
Demographic Characteristics (Confounders)			
**Race**			
Black	2 (11.1%)	16 (88.9%)	18 (12.4%)
Hispanic	12 (33.3%)	24 (66.7%)	36 (24.8%)
Multiracial/Other	5 (26.3%)	14 (73.7%)	19 (13.1%)
White	26 (36.1%)	46 (63.9%)	72 (49.7%)
**School Level ***			
High School	6 (18.2%)	27 (81.8%)	33 (22.8%)
Middle School	9 (19.6%)	47 (80.4%)	46 (31.7%)
Primary School	30 (45.5%)	36 (54.5%)	66 (45.5%)
**Sex**			
Females	21 (28.8%)	52 (71.2%)	73 (50.3%)
Males	24 (33.3%)	48 (66.7%)	72 (49.7%)
Income			
Median Family Income	$110,500	$100,000	$100,000
**Parental loss of income**			
Yes	6 (28.6%)	15 (71.4%)	21 (14.5%)
No	39 (31.5%)	85 (68.5%)	124 (85.5)
Health related attitudes and behaviors			
**Are parents concerned with COVID-19 vaccine safety for children? ***			
Not too concerned/not concerned at all	11 (55.0%)	9 (45.0%)	20 (13.8%)
Some concern/Very Concerned	27 (26.2%)	76 (73.8%)	103 (71.0%)
Not Sure	7 (31.8%)	15 (68.2%)	22 (15.2%)
**Does parent plan to vaccinate child against COVID-19**			
Yes	22 (33.3%)	44 (66.7%)	66 (45.5%)
No	3 (25%)	9 (75%)	12 (8.3%)
Not Sure	20 (29.9%)	47 (70.1%)	67 (46.2%)
**Parent working in medical setting**			
Yes	18 (37.5%)	30 (62.5%)	48 (33.1%)
No	27 (27.8%)	70 (72.2%)	97 (66.9%)
**Child receive the flu vaccine in the Fall of 2019 or Winter/Spring of 2020? ***			
Yes	40 (37.4%)	67 (62.6%)	107 (73.8%)
No	5 (13.2%)	33 (86.8%)	38 (26.2%)
**KAP and Unwanted Behaviors**	Mean (Std Dev)	Mean (Std Dev)	Mean (Std Dev)
Knowledge Score	15.00 (0.879)	14.55 (1.58)	14.69 (1.427)
Attitude Score **	6.96 (1.821)	8.07 (1.821)	7.72 (1.548)
Practice Score **	5.93 (1.176)	6.58 (0.997)	6.38 (1.093)
Unwanted Behavior Score	1.56 (1.439)	1.69 (1.398)	1.65 (1.407)
**Other Parental Characteristics**	Mean (Std Dev)	Mean (Std Dev)	Mean (Std Dev)
Healthy Days	25.13 (7.021)	24.53 (7.368)	24.72 (7.234)
Optimism Score	17.91 (4.486)	17.22 (4.620)	17.43 (4.575)
Resilience Score	3.17 (1.11)	3.07 (1.33)	3.11 (1.26)
**Child Participation in Sports**			
Yes	18 (29.5%)	43 (70.5%)	61 (42.1%)
No	27 (32.1%)	57 (67.9%)	84 (57.9%)
**Psychosocial health**			
Students At Risk for either Anxiety, Depression, or OCD	29 (34.9%)	54 (65.1%)	83 (57.2%)
Students At High Risk for either Anxiety, Depression, or OCD	7 (36.8%)	12 (63.2%)	19 (13.1%)
Anxiety-related symptoms (At Risk)	21 (34.4%)	40 (65.6%)	61 (42.1%)
Anxiety-related symptoms (High Risk)	5 (35.7%)	9 (64.3%)	14 (9.7%)
Depressive symptoms (At risk)	24 (36.9%)	41 (63.1%)	65 (44.8%)
Depressive symptoms (High risk)	1 (12.5%)	7 (87.5%)	8 (5.5%)
OCD-related symptoms (At Risk)	19 (31.7%)	41 (68.3%)	60 (41.4%)
OCD-related symptoms (High Risk)	6 (37.5%)	10 (62.5%)	16 (11%)

* *p* < 0.05 using a Chi-Square test to compare those attending school in-person vs. ERE. ** *p* < 0.05 using an Independent Sample *t*-test to compare those attending school in-person vs. ERE.

**Table 2 ijerph-18-08564-t002:** Nested Logistic Regression Models Examining the Effects of Sociodemographic Characteristics, Learning Environment, Parental Health Attitudes and Behaviors, and other Selected Factors on Students Presenting as At Risk for Anxiety.

Independent Variables	Model 1	Model 2	Model 3	Model 4	Model 5
**Race**	aOR (95% CI)	aOR (95% CI)	aOR (95% CI)	aOR (95% CI)	aOR (95% CI)
White (Reference)	--	--	--	--	--
Black	0.42 (0.12–1.43)	0.42 (0.12–1.47)	0.66 (0.16–2.72)	0.66 (0.16–2.71)	0.88 (0.2–3.82)
Hispanic	1.02 (0.42–2.51)	1.02 (0.42–2.52)	1.76 (0.61–5.09)	1.75 (0.6–5.08)	1.86 (0.62–5.6)
Multiracial	1.31 (0.42–4.08)	1.31 (0.42–4.14)	2.65 (0.64–11.07)	2.61 (0.62–11.03)	3.32 (0.74–14.82)
**School Level**					
High School (Reference)	--	--	--	--	--
Middle School	4.34 (1.44–13.07) *	4.34 (1.44–13.07) *	5.61 (1.63–19.26) *	5.59 (1.63–19.22) *	6.17 (1.74–21.9) *
Primary School	5.25 (1.87–14.75) *	5.22 (1.83–14.91) *	7.64 (2.24–25.98) *	7.66 (2.25–26.05) *	6.73 (1.92–23.51) *
**Sex**					
Male (Reference)	--	--	--	--	--
Female	1.83 (0.88–3.8)	1.83 (0.88–3.8)	2.11 (0.91–4.89)	2.14 (0.91–5.05)	2.24 (0.93–5.39)
**Income Level**					
Income above median (Reference)	--	--	--	--	--
Income below median	2.41 (0.82–7.12)	2.4 (0.81–7.13)	2.18 (0.59–8)	2.22 (0.59–8.27)	2.44 (0.63–9.51)
**Learning Environment**					
ERE (Reference)		--	--	--	--
In-Person		1.03 (0.46–2.28)	1.17 (0.42–3.27)	1.18 (0.42–3.28)	1.13 (0.39–3.26)
**Parental Health Attitudes and Behaviors**					
Attitude Score			1.36 (0.98–1.89)	1.37 (0.97–1.92)	1.33 (0.95–1.86)
Knowledge Score			1.62 (1.12–2.35) *	1.63 (1.12–2.36) *	1.71 (1.17–2.48) *
**Do you plan on having your child(ren) vaccinated against COVID-19 when a vaccine becomes available?**					
No (Reference)			--	--	--
Yes			5.71 (0.78–41.81)	5.69 (0.78–41.31)	5.47 (0.61–48.67)
Not Sure			3.98 (0.59–27.06)	3.92 (0.58–26.62)	3.31 (0.39–27.77)
Child receiving flu vaccine in 2019 **			1.29 (0.42–3.96)	1.28 (0.42–3.94)	1.55 (0.48–5.04)
Exercised less than before the pandemic **			1.91 (0.75–4.83)	1.93 (0.75–4.96)	1.64 (0.62–4.36)
Drank more alcohol than I did before the pandemic **			2 (0.7–5.7)	2.01 (0.7–5.74)	2.07 (0.71–6.07)
Ate more unhealthy food than I did before the pandemic **			0.9 (0.37–2.2)	0.91 (0.37–2.24)	0.84 (0.34–2.1)
Avoided going to the doctor for a non-COVID-19-related problem **			1.79 (0.69–4.63)	1.78 (0.69–4.62)	1.78 (0.69–4.62)
**Parental self-reported health**					
Healthy Days				1.01 (0.95–1.07)	1.01 (0.95–1.07)
**Is child practicing sports**					
No (Reference)					--
Yes					0.36 (0.14–0.93) *
**Fit (-2 Log Likelihood)**	177.421	177.417	145.724	145.697	141.129
**Constant**	0.141	0.140	0.000	0.000	0.000

* *p* < 0.05, ** The reference value for these variables is parents/children not engaging in the behaviors.

**Table 3 ijerph-18-08564-t003:** Nested Logistic Regression Models Examining the Effects of Sociodemographic Characteristics, Learning Environment, Parental Health Attitudes and Behaviors, and other Selected Factors on Students Presenting as At Risk for Depression.

Independent Variables	Model 1	Model 2	Model 3	Model 4	Model 5
**Race**	aOR (95% CI)	aOR (95% CI)	aOR (95% CI)	aOR (95% CI)	aOR (95% CI)
White (Reference)	--	--	--	--	--
Black	0.16 (0.04–0.68) *	0.17 (0.04–0.73) *	0.22 (0.04–1.1)	0.23 (0.05–1.12)	0.31 (0.06–1.59)
Hispanic	1.18 (0.47–2.94)	1.2 (0.48–3.01)	1.4 (0.52–3.75)	1.39 (0.52–3.72)	1.42 (0.51–3.96)
Multiracial	0.66 (0.21–2.1)	0.69 (0.21–2.21)	0.96 (0.26–3.56)	0.92 (0.25–3.47)	1.15 (0.29–4.61)
**School Level**					
High School (Reference)	--	--	--	--	--
Middle School	5.64 (1.83–17.41) *	5.66 (1.83–17.49) *	6.09 (1.76–21.14) *	6.09 (1.74–21.28) *	6.55 (1.79–23.98) *
Primary School	6.42 (2.23–18.48) *	6.09 (2.09–17.78) *	6.2 (1.89–20.36) *	6.24 (1.88–20.69) *	5.4 (1.56–18.66) *
**Sex**					
Male (Reference)	--	--	--	--	--
Female	2.27 (1.07–4.78) *	2.29 (1.08–4.85) *	2.84 (1.24–6.5) *	2.87 (1.24–6.62) *	2.99 (1.27–7.03) *
**Income Level**					
Income above median (Reference)	--	--	--	--	--
Income below median	1.75 (0.57–5.39)	1.69 (0.54–5.26)	0.94 (0.27–3.29)	0.96 (0.28–3.32)	1 (0.28–3.6)
**Learning Environment**					
ERE (Reference)		--	--	--	--
In-Person		1.27 (0.57–2.86)	2.18 (0.83–5.72)	2.31 (0.87–6.15)	2.35 (0.86–6.42)
**Parental Health Attitudes and behaviors**					
Attitude Score			1.3 (0.97–1.74)	1.3 (0.97–1.75)	1.28 (0.95–1.71)
Exercised less than before the pandemic **			2.17 (0.88–5.34)	2.29 (0.92–5.72)	1.88 (0.73–4.83)
Drank more alcohol than I did before the pandemic **			2.39 (0.85–6.67)	2.14 (0.75–6.16)	2.17 (0.74–6.37)
Ate more unhealthy food than I did before the pandemic **			1.51 (0.64–3.57)	1.55 (0.65–3.69)	1.46 (0.6–3.55)
Avoided going to the doctor for a non-COVID-19-related problem **			1.63 (0.62–4.25)	1.56 (0.6–4.08)	1.58 (0.59–4.22)
**Parent working Frontline**					
No (Reference)				--	--
Yes				0.64 (0.27–1.53)	0.78 (0.32–1.91)
**Is child practicing sports**					
No (Reference)					--
Yes					0.38 (0.15–0.93) *
**Fit (-2 Log Likelihood)**	170.683	170.345	149.868	148.835	144.245
**Constant**	0.145	0.135	0.006	0.006	0.010

* *p* < 0.05, ** The reference value for these variables is parents/children not engaging in the behaviors.

**Table 4 ijerph-18-08564-t004:** Nested Logistic Regression Models Examining the Effects of Sociodemographic Characteristics, Learning Environment, Parental Health Attitudes and Behaviors, and other Selected Factors on Students Presenting as At Risk for OCD.

Independent Variables	Model 1	Model 2	Model 3	Model 4	Model 5
**Race**	aOR (95% CI)	aOR (95% CI)	aOR (95% CI)	aOR (95% CI)	aOR (95% CI)
White (Reference)	--	--	--	--	--
Black	0.86 (0.29–2.56)	0.84 (0.28–2.57)	1.67 (0.42–6.57)	1.66 (0.42–6.57)	2.24 (0.54–9.26)
Hispanic	0.52 (0.21–1.26)	0.52 (0.21–1.26)	0.77 (0.26–2.32)	0.78 (0.26–2.38)	0.83 (0.26–2.67)
Multiracial	0.89 (0.3–2.64)	0.89 (0.3–2.64)	1.46 (0.37–5.73)	1.45 (0.36–5.81)	1.99 (0.47–8.39)
**School Level**					
High School (Reference)	--	--	--	--	--
Middle School	1.28 (0.49–3.35)	1.28 (0.49–3.35)	1.37 (0.42–4.43)	1.37 (0.42–4.46)	1.53 (0.45–5.21)
Primary School	1.7 (0.7–4.13)	1.72 (0.69–4.27)	1.78 (0.57–5.53)	1.79 (0.57–5.6)	1.45 (0.43–4.83)
**Sex**					
Male (Reference)	--	--	--	--	--
Female	1.34 (0.67–2.69)	1.34 (0.67–2.68)	1.56 (0.67–3.59)	1.57 (0.67–3.7)	1.64 (0.68–3.97)
**Income Level**					
Income above median (Reference)	--	--	--	--	--
Income below median	2.09 (0.76–5.73)	2.1 (0.76–5.8)	1.55 (0.43–5.58)	1.58 (0.43–5.84)	1.7 (0.44–6.51)
**Learning Environment**					
ERE (Reference)		--	--	--	--
In-Person		0.95 (0.44–2.06)	1.42 (0.49–4.11)	1.42 (0.49–4.11)	1.31 (0.43–3.94)
**Parental Health Attitudes and behaviors**					
Attitude Score			1.64 (1.14–2.37) *	1.65 (1.13–2.39) *	1.6 (1.1–2.34) *
Knowledge Score			1.29 (0.89–1.86)	1.29 (0.89–1.87)	1.42 (0.96–2.1)
**Do you plan on having your child(ren) vaccinated against COVID-19 when a vaccine becomes available?**					
No (Reference)			--	--	--
Yes			8.05 (0.75–86.6)	8.19 (0.73–91.45)	8.29 (0.56–121.78)
Not Sure			5.62 (0.55–57.39)	5.65 (0.53–59.9)	5.21 (0.37–73.43)
Child receiving flu vaccine in 2019 **			1.98 (0.63–6.28)	1.98 (0.62–6.3)	2.35 (0.7–7.96)
Exercised less than before the pandemic **			2.44 (0.96–6.2)	2.46 (0.96–6.3)	2.05 (0.77–5.49)
Drank more alcohol than I did before the pandemic **			1.93 (0.66–5.61)	1.96 (0.66–5.77)	2.16 (0.7–6.72)
Postponed vaccination for myself or my child **			1.35 (0.48–3.81)	1.35 (0.48–3.83)	1.65 (0.56–4.83)
Avoided going to the doctor for a non-COVID-19-related problem **			2.56 (0.98–6.66)	2.56 (0.98–6.66)	2.54 (0.96–6.72)
**Parental self-reported health/resilience**					
Healthy Days				1 (0.93–1.07)	1.02 (0.95–1.09)
Resilience Score				0.97 (0.65–1.45)	1.06 (0.7–1.62)
**Is child practicing sports**					
No (Reference)					--
Yes					0.31 (0.11–0.85) *
**Fit (-2 Log Likelihood)**	189.717	189.699	144.162	144.137	138.606
**Constant**	0.480	0.487	0.000	0.000	0.000

* *p* < 0.05, ** The reference value for these variables is parents/children not engaging in the behaviors.

## Data Availability

Data may be requested from the PI of the project Sarah L. McKune or the corresponding author of this manuscript once it is released. The complete unidentified data set will be released six months after the completion of the longitudinal study (estimated date, October 2022).

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
