# Peer review of "Psychosocial Health of K-12 Students Engaged in Emergency Remote Education and In-Person Schooling: A Cross-Sectional Study"

_ijerph, 2021, doi:10.3390/ijerph18168564_

Round 1

Reviewer 1 Report

This study examined differences in mental health between K-12 students who engaged in remote learning during the COVID-19 pandemic and those who attended in-person classes. The topic is relevant and contributes to our knowledge of learning modality and mental health among school-age children during the COVID-19 pandemic. There are some suggestions to the authors that could help strengthen the manuscript; these are detailed below.

In the abstract: Since the survey was cross-sectional, the term “predictive” should be avoided. Instead, the authors could use “associated with” instead.

Page 2, there are a few typos: 1) lines 71-72 there be a space between “online” and “[13”]; 2) lines 73-74 day should be plural.

Page 3, there is a typo in lines 116-117 (“the questionnaire included parental or self-reported of symptoms” perhaps this should read “parental or self-report of symptoms”)

In some places of the manuscript, the authors refer to COVID-19 as SARS-CoV-2, in others COVID-19. I would suggest a definition at the beginning and then consistent terminology throughout.

What was the rationale for creating the “no risk,” “low risk,” and “high risk” constructs? Has this been done in prior literature? How many items were part of this scale/measure? What was Cronbach’s alpha for the scale?

What was the rationale for not separating/conducting a sub analyses of those in traditional in-person school (e.g., elementary students) from those in the pod-style in-person (e.g., middle/high school students)?

Was a power analysis conducted to ensure sufficient power/sample size?

What is parental KAP (mentioned on p. 4)?

The finding that parental knowledge about COVID-19 is positively associated with anxiety may need to be further explored; it is possible that parental education level could impact perceived child mental health. Moreover, parents with more realistic knowledge and expectations about COVID-19 could possibly be more attuned to their children’s mental health and well-being.  

Author Response

Thank you for your comments. Please see the attachment for the responses to each point. Our responses are italicized. Please let us know if we have addressed your concerns.

Reviewer 2 Report

A question which surely must be asked is addressed and quite well explored. To strengthen the paper please consider the following:

  • The presentation of all the results tables may not be necessary. The full set of results may be more appropriate in an appendix. Consider how (if) the 'main messages' of the results can be more effectively presented.  Can model drawings be used?
  • In limitations, consider also mentioning the variation in characteristics of the online learning students experienced. Although it is outside scope of the presented research to capture the nature of the learning provided by teachers in the courses, I believe there should be some mention of this as a potentially important factor (as an educator myself, it is evident that how the teacher interacts with students makes a difference to the mental health of students).
  • In research papers, my own view is that phrases such as "for a variety of reasons" (line 329) should be avoided or at the least should state some potential reasons (perhaps by referring to other literature which supports the authors' view about the reasons)

  • Some aspects of the discussion perhaps require a little more depth and analysis. In stating that the more the parent knows about.. the more...- how was the quality of knowledge of the parent assessed? Best to make this clear.

Otherwise, an informative paper overall. Main area for improvement is to more effectively and succinctly communicate the important findings and recommendations.

Author Response

(The authors gave the same response as above.)
